# A Physiological-Signal-Based Thermal Sensation Model for Indoor Environment Thermal Comfort Evaluation

**DOI:** 10.3390/ijerph19127292

**Published:** 2022-06-14

**Authors:** Shih-Lung Pao, Shin-Yu Wu, Jing-Min Liang, Ing-Jer Huang, Lan-Yuen Guo, Wen-Lan Wu, Yang-Guang Liu, Shy-Her Nian

**Affiliations:** 1Department of Computer Science and Engineering, National Sun Yat-sen University, Kaohsiung 80424, Taiwan; sjpao@esl.cse.nsysu.edu.tw (S.-L.P.); hywu@esl.cse.nsysu.edu.tw (S.-Y.W.); 2Department of Sports Medicine, Kaohsiung Medical University, Kaohsiung 80708, Taiwan; taiga1115@gmail.com (J.-M.L.); wenlanwu@kmu.edu.tw (W.-L.W.); 3Digital Content and Multimedia Technology Research Center, National Sun Yat-sen University, Kaohsiung 80424, Taiwan; 4Department of Medical Research, Kaohsiung Medical University Hospital, Kaohsiung 80708, Taiwan; 5College of Humanities and Social Sciences, National Pingtung University of Science and Technology, Pingtung 91201, Taiwan; 6Green Energy & Environmental Laboratories, Industrial Technology Research Institute, Hsinchu 31040, Taiwan; ygliu@itri.org.tw (Y.-G.L.); shnian@itri.org.tw (S.-H.N.)

**Keywords:** thermal sensation, thermal comfort, PMV (predicted mean vote), sensation modeling, personalized thermal comfort strategy, EMG, ECG, EEG, GSR, body temperature

## Abstract

Traditional heating, ventilation, and air conditioning (HVAC) control systems rely mostly on static models, such as Fanger’s predicted mean vote (PMV) to predict human thermal comfort in indoor environments. Such models consider environmental parameters, such as room temperature, humidity, etc., and indirect human factors, such as metabolic rate, clothing, etc., which do not necessarily reflect the actual human thermal comfort. Therefore, as electronic sensor devices have become widely used, we propose to develop a thermal sensation (TS) model that takes in humans’ physiological signals for consideration in addition to the environment parameters. We conduct climate chamber experiments to collect physiological signals and personal TS under different environments. The collected physiological signals are ECG, EEG, EMG, GSR, and body temperatures. As a preliminary study, we conducted experiments on young subjects under static behaviors by controlling the room temperature, fan speed, and humidity. The results show that our physiological-signal-based TS model performs much better than the PMV model, with average RMSEs 0.75 vs. 1.07 (lower is better) and R^2^ 0.77 vs. 0.43 (higher is better), respectively, meaning that our model prediction has higher accuracy and better explainability. The experiments also ranked the importance of physiological signals (as EMG, body temperature, ECG, and EEG, in descending order) so they can be selectively adopted according to the feasibility of signal collection in different application scenarios. This study demonstrates the usefulness of physiological signals in TS prediction and motivates further thorough research on wider scenarios, such as ages, health condition, static/motion/sports behaviors, etc.

## 1. Introduction

### 1.1. Motivation and Objective

The purpose of heating, ventilation, and air conditioning (HVAC) systems is to provide comfortable thermal environments for occupants. The key problem is to predict occupants’ thermal sensation (TS). Nowadays, the models for such prediction are mostly static, with Fanger’s predicted mean vote (PMV) model [1] being a widely used one. It has been adopted in ASHRAE 55 [2] and ISO 7730 [3] international standards for evaluating indoor TS.

PMV model predicts the thermal comfort based on environmental parameters, such as temperature, air velocity, humidity, etc. In addition, PMV also takes into account the occupant’s personal conditions, such as the clothing insulation and metabolic rate. However, it is difficult to obtain a person’s actual metabolic rate in real time and, thus, a representative value consulted from a metabolic rate database is often used. Therefore, it is difficult to use PMV in real time.

As electronic sensors have become easily accessible, it is now possible to measure the human body’s physiological signals in real time and use them for accurate HVAC control. Therefore, we propose to develop a thermal sensation model which adopts occupants’ real-time physiological signals as an indication of their body status, in addition to the environmental parameters. We conducted a climate chamber experiment to collect subjects’ physiological signals and TSs under different thermal conditions. There are five physiological signals, ECG, EEG, EMG, GSR, and body temperature, measured. As a preliminary study, we conducted experiments on young subjects under static behaviors by controlling the room temperature, fan speed, and humidity. We modeled TS by 19 extracted physiological features and 3 environmental features. The relationship between TS prediction and the physiological signals is examined by the proposed modeling process. The physiological signals were ranked to show the degree of their contributions to the TS prediction. Finally, we will compare our proposed model with PMV model.

### 1.2. Current State of Research

Thermal comfort is closely related to the wellbeing of human beings. Uncomfortable environment may cause a decline in work efficiency and occupants’ physical discomfort. HVAC systems were invented and massively utilized in buildings. The energy consumption for thermal adjustment in business buildings is around 50% of the total energy consumption [4]. The thermal comfort of the environment not only affects the occupants’ work quality, but the sustainability of the buildings [5].

Novel research of HVAC system focuses not only from the perspective of energy efficiency, but also on how they act with occupants’ thermal comfort. Xie et al. [6] examined the design factors of indoor radiant heating system and discovered that water temperature within the radiator heavily affects thermal comfort. Oh et al. [7] examined the influence of mist-spraying systems and proved the systems caused decreases in air temperature and increases in thermal comfort in hot weather. Zhang et al. [8] found the effect of solar radiant to the indoor environment is correlated with window’s characteristics.

Thermal sensation is the occupants’ perceptual response to the thermal environment, and a thermal sensation model is a simulation model to predict how occupants will respond to the thermal environment. The pioneer study of thermal sensation model is Fanger’s PMV model [1]. The concept of PMV is about thermal balance. Four environment parameters (air temperature, air velocity, mean radiant temperature, and relative humidity) and two body-related parameters (clothing and metabolism rate) are included in the PMV formula to estimate the average TS of the occupants. PMV as a widely used TS index has been included as an indoor environment quality metric in several indoor environment standards, such as ISO 7730 [3] and ASHRAE Standard 55 [2]. In field applications, PMV model predicts the range of acceptable temperature, airspeed, and relative humidity by considering target thermal comfort level and the clothing insulation and metabolic rate of occupants.

Since stricter control of the thermal environment does not necessarily lead to a more comfortable thermal environment [9], the accuracy of TS prediction is crucial in HVAC control. PMV is not suitable for real-time operation, since it is difficult to obtain a person’s actual metabolic rate in real time and, thus, a representative value consulted from a metabolic rate database is often used. Nicol and Humphreys [5] mentioned PMV does not outperform simple indices, such as air temperature in the field study, and non-real-time adjustment may decrease occupant thermal comfort.

As an index according to the statistical result of experimental data over time, PMV sometimes produces increasing errors over time. The range of temperature in ASHRAE suggested the winter comfort zone has risen year after year since the 1940s [9]. It shows the habits of human beings changed as the climate, culture, and technology changed.

The adaptation of occupants would degrade the accuracy of PMV model. In experiments, occupants might get used to the experimental field and lead to different response in TS. De Dear and Brager [10] proposed an adaptive model and pointed out one’s TS will be influenced by long-term outdoor climate. Wu et al. [11] found the adaptive model shows better work than PMV model in thermal acceptance, thus reduce energy consumption in summer with fewer AC operations. Albatayneh et al. [12] recorded the free-run indoor environment change and calculated the equivalent necessary AC operations, and found the adaptive model significantly reduces the time needed for heating and cooling. Soebarto and Bennetts [13] found higher temperature is more tolerable for warm climate residents than the ones who lived in a cold climate. It indicates that thermal sensation could be affected by personal characteristics, such as cold syndrome, daily routine, disability, and state of health [14].

It should be noted that thermal sensation models are restricted by their assumptions on the environments, e.g., PMV performs better with AC environment and the adaptive model is limited with free-run buildings [12]. Whether rebuilding TS models under different scenarios or developing methods to adjust existed TS models, enormous work is required. To avoid tremendous beforehand investigation on historical area climate and occupants’ behavior, an alternative approach is to model TS with physiological signals which can state one’s body status. Li et al. [15] first modeled occupants’ thermal sensation by the physiological signals, i.e., skin temperature and heart rate, measured by a wristband. An HVAC control system was then built according to the thermal sensation model and reached a more comfortable environmental conditioning and reduced 13.8% daily energy consumption [16]. Mohammad et al. [17] used a wristband to obtain real-time metabolic rates and enhance PMV calculation. Deng and Chen [18] developed an HVAC control system based on the physiological features measured by a wristband, i.e., skin temperature, skin humidity, and heart rate. The result shows that less than 5% of occupants feel discomfort under the system. The limitation is that the wristband measures only heart rate and wrist temperature.

Thermal sensation is the consequence of complex interaction between thermoregulation and cognition. Takahashi et al. [19] indicate thermoregulation includes several physiological activities, such as vasodilation, vasoconstriction, shivering, and sweating; hence, the related physiological signals could possibly be the evidence of TS.

## 2. Proposed Methods

The objective of this study is to develop a TS model which can make TS predictions by using physiological signals to reflect occupants’ physiological status in real time. To develop the target model, we conduct a chamber experiment to collect the required data: environmental and physiological signals under different environmental conditions. The collected environmental and physiological signals are then used to infer the heat sensation of each subject. The experiments are conducted in task-oriented environment-controllable rooms. To make the experiment go smoothly, we have developed experiment assistant tools to deal with environmental control and monitoring and physiological signal sensing. After the experiments, based on the information obtained, several regression models are selected for TS modeling.

### 2.1. Experiment Field

We prepared two rooms, the preparation room and the experiment room, as the experiment sites (as shown in Figure 1). Both rooms were about 26.5 m^2^. The preparation room was used to bring the subject’s physiological state back to a baseline value. The equipment and decoration of the two rooms are the same. To make the subjects feel relaxed during the experiment, wood-grain wallpaper and flooring were used, and daylight color lighting was chosen, following Zhang et al. [20]. The desk and the seat position of subjects were in the center of the room. The air conditioner was about 3 m high on the wall and in front of the desk, and its wind direction was downwards to avoid direct impact to the subjects. The rooms had no window and only one door as a connection to outside. The nearest window near the rooms was about 20 m from the entrance of the rooms; thus, solar radiation could be considered minimized and no direct impact to the experiments.

### 2.2. Experiment Assistant Tools

The experiment assistant tools (as shown in Figure 2) have been built to assist the staff to control the equipment and integrate data gathered from various sensors. The experiment assistant tools consist of three subsystems, i.e., (1) an environment measurement system to record the environmental conditions, such as air temperature, air speed, and relative humidity, that a subject was exposed to, (2) an environment control system for staff to easily control the AC system, including the AC and the ceiling fan, and (3) a physiological signal measurement system to record all physiological signals simultaneously. The information of (1) and (2) were integrated into a GUI, so the control of the experiment environment and the measurement could be synchronized.

### 2.3. Experiment of Physiological Signal Measurement under Different Environmental Conditions

The experiment is shown in Figure 3a. The environment was adjusted by the staff before the experiment. In the first 5 min (setup stage), the experiment procedure was explained to the subject and physiological signal sensors were installed on the subject. The next 15 min were the preparation stage, whereby the subject rested in the preparation room to reach a baseline physiological state. Following the 15-min experiment stage, a physiological signal measurement was performed in the experiment room. Since Ji et al. [21] indicate skin temperature will settle down to a relatively stable state after environment change within 10 min, we allocated 15 min for this stage. The physiological features were extracted from the last 5 min of the records. During the experiment stage, the subject was asked to watch a peaceful educational video to simulate an office scenario and to avoid emotional stimulation. At the end of the process, the subject was asked to fill out a thermal comfort questionnaire to record their TSs and comfort levels at the time.

In each trial of the experiment, the experiment room was set to different environmental conditions, while the preparation room was maintained the same. Subjects with similar growing-up locations have similar thermal perceptions [9]. Since the subjects were healthy Taiwanese residents, the experiment environment was set to be cold or hot for typical Taiwanese residents. We set the preparation room as 25 °C, 70% relative humidity (RH), and a medium speed ceiling fan. The experiment room had eight different environment settings, which were formed by two kinds of temperature (23 °C and 27 °C), two kinds of RH (60% and 80%), and two kinds of wind speed (breeze and strong wind of the ceiling fan), as shown in Figure 3b.

The order of the environment settings was randomized as much as possible; the subjects chose their own convenient time slots from our experiment schedule.

### 2.4. Measured Signals

We measured electroencephalography (EEG), electrocardiography (ECG), body temperature, electromyography (EMG), and galvanic skin response (GSR). Figure 3c shows the location of the measured signals:ECG: a single lead signal is measured across the chest, with the reference electrode placed at the lower edge of the left costa.EEG: measurement site is the prefrontal lobe, with the electrodes placed at FP1 and FP2 of the international 10–20 system, and the reference electrode placed behind the right ear.EMG: measured from the gastrocnemius of the left calf, and the reference electrode is placed on the knee.GSR: measured from the index finger and the middle finger of the left hand.Body temperature: Yao et al. [22] found the extremity body temperature is more susceptible to environmental influences than the core body temperature. Body temperatures are measured from the left arm, left chest, and left calf as representations for core and extremity body temperatures.

### 2.5. Subjects

We selected 20 healthy young adults, aged 24.1 ± 3.6 years, with a height of 167.7 ± 8.7 cm and a weight of 69.3 ± 20.1 kg. The male-to-female ratio of these 20 subjects was 9:11. Each subject had a meal before the experiment. Coffee, tea, and alcoholic drinks were prohibited to avoid complex physiological influences.

### 2.6. IRB

This measurement was regulated by the Institutional Review Board (IRB) of Kaohsiung Medical University with the number KMHIRB-E(I)-20200266. Each subject was informed of the entire test procedure and signed the participant consent form and agreed that the data measured in the experiment could be used for research purposes. In terms of dress, we required subjects to wear short-sleeved shirts and shorts, and no additional warm clothing, such as jackets, during the experiment.

### 2.7. Quality Filtering of Data

Some of the collected physiological signals are too noisy to be used. The collected signals were analyzed to eliminate the low-quality samples to ensure the reliability of the subsequent steps.

We removed the samples with serious noise. The total number of samples obtained from the experiments was 160, and the number of samples that could be used after removing the damaged samples was 90. The main reason for removing a larger number of samples is that one impaired signal, e.g., EEG, would void all the sample data of the subject under the corresponding room conditions.

### 2.8. Feature Extraction

We then extracted 19 physiological features from the 5 measured physiological signals and 3 environmental features for regression analyses. Table 1 shows the extracted features and their explanation.

### 2.9. Modeling

The modeling methods we used were linear regression, Gaussian process regression, SVM regression, and decision tree. We used MATLAB R2020b and its application Regression Learner for modeling as the signal process platform.

Due to the large number of extracted features, i.e., 22 physiological and environmental features, it was not possible to try all combinations of features for modeling due to the time constraint. Forward selection and backward elimination algorithms are commonly used in multiple feature modeling. Forward selection starts from an empty set and adds one feature at a time according to the feature’s correlation. If the added feature contributes to the model performance, then it is retained, and vice versa. Backward elimination begins from the combination of all features and removes one feature at a time according to the correlation of each feature. If the removal benefits the model performance, then the selected feature is eliminated from the combination. These methods do not consider the effect of combining the features which are not contributing to the performance. We use the heuristic algorithm to select the features used in the model.

The proposed feature selection algorithm contains two stages: feature increment and redundant feature removal, as shown in Figure 4. In the feature increment process, as depicted in Figure 4a, we start from setting the overall best combination S_best_ and temporal best combination T_best_ as empty sets. In each round, we add one feature in the unselected feature set S_uc_ to T_best_ to form a new combination T_i_. The collection of all possible T_i_ forms C_new_. Then, we build a set of TS models M(C_new_) for C_new_, and evaluate its performance P(M(C_new_)). T_best_ will be updated to the best T_i_ and the performance will be compared to the current S_best_. If T_best_ performs better than S_best_, then S_best_ will be updated to T_best_ and begin a new round of feature increment. Once T_best_ performs worse than S_best_, the feature increment process will be ended. Then, the S_best_ will be passed into the redundant feature removal process as in Figure 4b. At the beginning, we set T_best_ as S_best_, and form C_new_ by removing every feature in T_best_ one by one. Similar to the feature increment process, we model M(C_new_), evaluate performance P(M(C_new_)), update T_best_, and compare the P(M(T_best_)) and P(M(S_best_)). If P(M(T_best_)) is better than P(M(S_best_)), then S_best_ will be updated to T_best_. If T_best_ performs worse than S_best_, then the count of worse T_best_ since the last time S_best_ was updated, Time_worse_(after S_best_ update), will be incremented. If it is not the first time that T_best_ performs worse than S_best_, the redundant feature removal process will be ended. Otherwise, another round of redundant feature removal starts from forming new C_new_ based on the current T_best_. The result of S_best_ and its model M(S_best_) are the final answer.

### 2.10. Statistical Analysis

To make a preliminary estimation of the relationship between the measured physiological signals and TS, we conducted an analysis of Pearson’s correlation between each feature and TS. The analysis was carried out using the statistical software SPSS with the selected 90 samples, including 50 samples of male and 40 samples of female subjects.

## 3. Results

### 3.1. Correlation Analysis of Thermal Sensation (TS)

The results of the correlation analysis are shown in Table 2. Of all 19 features, there are 13 features, marked with * or **, which show statistical significance (*p* < 0.05) and with a moderate to weak correlation coefficient (0.558 through 0.168).

The results show that the most prominent features are EMG features (EMG_IEMG, EMG_MAV, EMG_RMS, and EMG_SSI) and body temperatures. On the other hand, the beta wave and average of EEG are also prominent.

As for the environment features, both the room temperature and wind velocity are significant. The relative humidity, measurable under typical AC room conditions, is not significant.

### 3.2. Modeling Results

#### 3.2.1. Best Model

The proposed feature selection algorithm explored 92 feature combinations. The best model used Gaussian process regression (GPR).

The features used in the best model are shown in Table 3. The model performance of the best feature combination can reach RMSE 0.807 and R^2^ 0.75. During the modeling process, it was found that RH does not help performance. Perhaps the RH setting points in the experiment were typical to Taiwan residents and the difference between setting points of RH was not enough to impact the TS.

The best feature combination comes from four physiological signals, EMG, ECG, EEG, and body temperature. We found the effect of EMG was particularly strong, and all EMG features were included in the model. This coincides with the findings of Yao et al. [22] and Sollers et al. [23]; they mentioned that ECG and EEG are related to thermal comfort. We further conducted experiments to quantify the relationships. On the other hand, body temperature, as an important characteristic of a warm-blooded animal, shows its importance within the feature selection sequence of modeling.

#### 3.2.2. Comparison with PMV

To evaluate the performance of the selected feature combination in assessing TS, PMV was used as the comparison object. The PMV data were calculated from the CBE Thermal Comfort Tool [24] provided by the U.C., Berkeley, CA, USA. The measured data were used for air temperature, humidity, and wind speed, while the radiation temperature was assumed to be equal to the air temperature with reference to Matzarakis and Amelung [25], since the experiment rooms are considered to be uniform. There are two components of personal factors, the human metabolic rate and clothing. We follow the ASHRAE-55 standard provided compliance table for various activities, with 1.0 met for seating and 0.5 clo for short-sleeved shirts and shorts as the set parameters. To avoid overlap between the training data and test data, we randomly sampled 72 data as the training set and the remaining 18 data as the test set. A total of three trial tests were conducted, with the test set and training set resampled each time. Each time, the model was retrained using the best feature combination and the Gaussian process regression model.

The results are shown in Table 4. In the three randomized 18 data, the R^2^ of the model using physiological signals can reach above 0.7. The results of the three trials of modeling showed an increase in R^2^ of at least 0.13 and a decrease in RMSE of at least 25.7% compared to PMV, indicating that our proposed model, with higher explainability and lower error, is better than the PMV model. This result demonstrates the feasibility of using physiological signals to assist in the assessment of thermal comfort.

#### 3.2.3. Examination on Gender

To evaluate whether gender will influence the performance of the proposed model, we examine the TS prediction error of male and female subjects. We randomly select 40 male samples and 32 female samples as the training set, and the remaining 10 male samples and 8 female samples are used as the test set. We intend to make the proportion of males to females the same in the original sample set, training set, and test set. The model was retrained with the training set, followed by the examination of the model’s TS prediction error on test set. We performed independent sample t-test and Mann–Whitney U test to examine the statistical difference in TS prediction error between male samples and female samples within the test set. Independent sample t-test is commonly used on two independent groups to determine whether there is any statistical evidence indicating that the means of two groups are statistically different. Mann–Whitney U test is very similar to the t-test but it examines the median of two groups and is specific for nonparametric statistics. The RMSE of TS prediction error on the whole test set, male samples, and female samples is 1.065, 1.092, and 1.03, respectively. Independent sample t-test and Mann–Whitney U test are applied on the TS prediction error of each gender. The result of t-test and Mann–Whitney U test are shown in Table 5 and Table 6. The *p*-value of both tests are geater than 0.05, indicating that the model errors of male subjects and of female subjects are not statistically significantly different from each other. This shows our model is applicable to both male and female genders.

#### 3.2.4. Physiological Signal Ranking

In the diverse real world, not all physiological signals are available; their measurement costs are different. Therefore, we want to rank the importance of physiological signals in assessing TS.

The process of ranking the importance of physiological signals is similar to the first stage of the modeling algorithm but operates on signal-wise feature sets. The signal-wise feature sets contain all features from a single physiological signal within the best feature combination. It starts from the combination with only environmental features. In each round, one best performed feature set is added until all signal-wise feature sets are tried. The order of physiological signal addition is shown in Table 7. Since body temperature is a physiological signal that can be measured more easily, we include this combination into the table.

If only one physiological signal can be selected, the easiest signal is body temperature. It is simple but, indeed, improves the performance. However, the even better one is EMG, which has better performance than body temperature. If two signals can be considered, EMG and body temperature could be selected. If three signals can be considered, ECG can be included. Finally, if four signals can be considered, then EEG can also be included, resulting in the best performance (RMSE = 0.807, R^2^ = 0.75). The order can be used as a reference for selecting the types of physiological signals under different application scenarios.

## 4. Discussion

This study tried to include physiological signals which are possibly related to thermoregulation. Besides widely used body temperature, the result shows EMG, EEG, and ECG are also able to contribute to TS prediction. This demonstrates the possibility of obtaining personal sensations from physiological signals. The proposed TS model predicts TS by using environmental and physiological signals and can provide a real-time prediction of occupants’ demand. By contrast, most of the earlier TS models need to estimate occupants’ conditions beforehand, e.g., activity level and clothing insulation, and make them unable to respond to occupants’ body status in real time.

The proposed model outperforms the PMV model. The proposed model gives an average RMSE of 0.75 and R^2^ of 0.76, while the widely used PMV model’s ones are 1.07 and 0.43, respectively, in our study case. Koelblen et al. [26] collected data from previous studies to validate several TS models, and the validation result of PMV shows the TS prediction RMSEs of databases are ranged from 0.2 to 1, implicitly indicating that different measurement scenarios lead to different prediction errors of the PMV model. A possible reason for varied RMSEs is lacking consideration to long-term climate change. Humphreys and Nicol [5] indicate that PMV may bias according to mean outdoor air temperature instead of daily maximum or minimum. Another reason for the RMSE difference might have been caused by our radiant temperature assumption. To lower the burden of data acquisition, we assumed radiant temperature is equal to the indoor air temperature. The difference between the two temperatures can cause the PMV model to perform worse. From the perspective of information collection, the result still shows that, under the scenario of lacking radiant temperature, the proposed model’s performance is better than the PMV model. Besides comparison with the PMV model, the effect of physiological features on TS prediction still can be found with our modeling procedure. During the procedure, the models with both physiological and environmental features outperform the ones with only environmental features.

Among the adopted signals of the proposed model, it shows EMG from the calf is the most important factor. Studies of EMG show that activity of EMG will be influenced by exposure temperature. Bell [27] examined the relationship between EMG amplitude and force from rectus femoris under different exposure temperature and found EMG amplitude increases while exposure temperature decreases with a fixed force given. Bell et al. [28] recorded the shivering EMG signals from six muscle groups under a long exposure of cold environment and found that EMG under shivering increased on both central and peripheral muscles. Our modeling result also shows EMG could be an important factor to estimate impacts from environment. A possible reason is the experiment conditions in this study made subjects tend to feel cool or cold, causing subjects to make more small movements, e.g., shivering, posture adjustment, or tapping their feet. EMG in static scenarios, e.g., the difference in sedentary EMG between different temperatures, can be verified in the future.

Body temperature shows secondary importance. In previous studies, each part of the body shows different fluctuation to the change in environment temperature. Body parts also differ in thermal sensitivity. There are some studies that describe the relationship between temperatures from several body parts and TS. Yao et al. [22] analyzed the relationship between thermal comfort and body temperature and discovered both local and mean skin temperatures are sensitive to environment temperature. Zhang et al. [29] use local body temperatures as indices of modeling thermal sensation and use the obtained thermal sensations to model thermal comfort in the following research [30]. Our model includes the temperatures from the chest and calf. The temperature from the chest stands for the core temperature, which undulates less than the ones from other body parts and may be the representative of the daily body temperature of a subject. The calf’s temperature presents the easy-affected limb temperature. Perhaps the exposure level of wearing shorts made the combination of temperatures from the chest and calf perform the best.

The third and fourth important signals are EEG from the forehead and ECG from the chest. EEG from the forehead can judge whether the subject is relaxed or focused. The result shows the beta band from EEG affects TS the most. The power of the beta band can be used to score one’s attention level. It is possibly because a sufficiently low temperature is helpful for being focused over a short period of time [31]. The features adopted from the ECG are heart rate variability (HRV) features, i.e., SDNN and LF/HF. HRV is related to the activity of the sympathetic and parasympathetic nervous systems [32,33], and can be used as an index of body regulation. Choi et al. [34] verified the potential heart rate and its variation as indices of the thermal comfort model under different activity levels and environment temperatures. The result shows the variation in heart rate is more representative than the heart rate itself. Our model also shows a similar result. The reason ECG features are not highly important in the proposed model is possibly because the relationship between the HRV features and TS is in the shape of a “smile curve”. LF/HF will be low when TS is neutral and be high when TS is hot or cold [22]. With different ambient temperature in the experiment, Zhu et al. [35] also found similar result on the LF/HF against TS curve. The linear GRP method of our model is unable to respond to the nonlinear relationship. In the future, more complex models, such as deep learning, can be adopted for TS modeling.

In this work, GSR did not show importance in TS modeling, and can be regarded as the consequence of a sedentary scenario. In the results of Gerrett et al. [36] and Xu et al. [37], GSR was highly correlated with TS in sporting scenarios. For the generality of the TS model, the sporting scenarios should be considered and GSR should be further discussed in future works.

Gender is a factor that influences one’s thermal sensation. To evaluate the effect of gender on the proposed model, we test the error of thermal sensation prediction on male and female groups by using t-test and Mann–Whitney U test. The RMSE of TS prediction error on the whole test set, male samples, and female samples is 1.065, 1.092, and 1.03, respectively. Although the RMSEs of two genders are a little bit different from each other, the result of both tests shows no statistically significant difference between two genders on TS prediction error. A possible reason is that the physiological signals already contain information about the genders; thus, the proposed model is applicable to both male and female genders. The PMV model also contains similar characteristics. Although PMV model does not include gender, it adopts metabolic rate, which can be affected by gender, as an input and, thus, need not additionally consider genders. Al-Mallah et al. [38] found males have a higher metabolic rate than females who are in their middle age. Sabounchi et al. [39] also show that gender is an important factor in Basel metabolic rate prediction. Another possible reason is that gender may not be as effective on thermal sensation as expected. Wang et al. [14] conducted a review of the studies on thermal comfort and found no consistent conclusions on the efficacy of gender. Further research on the relationship between gender and thermal sensation could be conducted in the future.

The difference in thermal history may lead to a different result from this study. The experiment in this work was conducted at Kaohsiung, Taiwan, and all subjects were healthy residents in Taiwan. According to Gautam et al. [40], thermal history affects one’s thermal preference. Chen et al. [41] discovered that people who lived in severe cold area feel comfortable with the thermal sensation ranging from “slightly cool” to “hot”. Subjects from other countries might lead to different conclusions due to their thermal history, body shape, and culture differences.

Physiological signals other than the chosen ones in our work may also have a potential on thermal sensation assessment. Oi et al. [42] tried to include brain fMRI images as a reference of TS estimation. fMRI images contain more spatial information of brain activities, while the single-point EEG signal which we used contains more temporal information. For long-term applications, compared with fMRI, EEG is a more suitable target due to its convenience. Arens et al. [43] proposed a TS model by integrating TS from body parts. It is possible to model TSs of different body parts with different models and merge into the overall TS; each model could be modeled by adopting different physiological signals. Kingma et al. [44] proposed a method of using the changes caused by nerve conduction to evaluate TS. In the study, by verifying with history records, neurophysiology showed its effectiveness in TS prediction. Neurophysiology methods can also be adopted in future studies after careful consideration.

The fluctuation of environmental conditions could possibly introduce errors into this study. Due to the limitation of the consumer-grade AC system, the environmental conditions were not as stable as expected. The uniformity of the environment might not be as expected consequently. Green [45] and Fang et al. [46] indexed that overall TS could be affected by the TSs of body parts, which may be varied by the influence of nonuniform environment. However, Arens et al. [43,47] showed overall TS is less sensitive than the TSs of body parts. Although errors in uniformity might cause fluctuation in the TSs of body parts, it may not overly bias the overall TS. The effect of environment control errors could be further discussed in future studies, while the difference between experiment and field studies could be researched at the same time.

The resolution on the environment control points might affect the accuracy of the proposed model. Because of the limitation in experiment ability, we chose two setting points on each environment condition, forming a total of eight environment settings. It is possible to choose fewer environment conditions, e.g., air temperature and air velocity, which are showing their importance in this study and increase the setting points on each condition. This could enhance condition resolution and verify whether the linear relationship between TS and environmental conditions persists.

The design of the questionnaire and subjects’ understanding of it may introduce errors. The questionnaire in this study adopts ASHRAE 7-point TS scale. Humphreys et al. [48] indicate same TS values from different subjects could be nonequivalent according to the different understanding of each scale’s description. This could lead to errors in linear regression. In the future, whether it would cause a drop in the model’s performance could be further researched, or nonlinear modeling methods might be considered as replacements.

## 5. Conclusions

This study proposed a thermal sensation (TS) prediction model using physiological and environmental features. By adopting physiological features as indications of body status, TS prediction becomes more accurate and it is possible to measure response to occupants’ personal feeling in real time. The results show that our physiological-signal-based TS model outperforms the PMV model, with average RMSEs 0.75 vs. 1.07 (lower is better) and R^2^ 0.77 vs. 0.43 (higher is better), respectively, indicating our model prediction has higher accuracy and better explainability. We also ranked the importance of adopted physiological signals by their efficacy of TS prediction in descending order as follows: EMG, body temperature, ECG, and EEG. In real-world applications, because not all the physiological signals are available or the budgets are limited, a proper subset of physiological signals can be selected from our proposed rank list. The physiological signals we adopted can be easily measured by wearable sensing devices, making them feasible in a wide range of application scenarios. The proposed model can be used not only for air conditioning, but also for assessing the TS needs of nonverbal subjects, such as infants or bedridden patients, as a guide to environment adjustment.

Future research can extend and explore several points: (1) reducing measurement noise to produce more usable samples, (2) applying more environmental set points and scenarios of different activity to increase the applicability of the model, (3) conducting measurements on different groups of subjects and study their TS similarity and differences, (4) the relationship between physiological signals and TS may be nonlinear, so more complex deep learning models can be used for modeling and prediction, and (5) incorporating physiological signals into future intelligent AC control systems.

## Figures and Tables

**Figure 1 ijerph-19-07292-f001:**
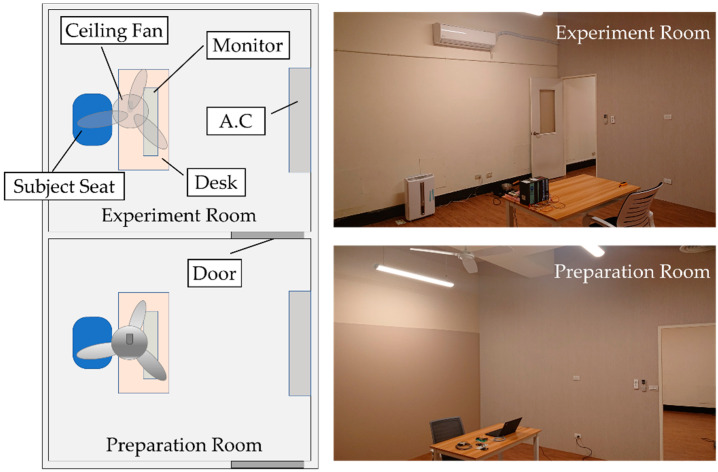
Experimental rooms.

**Figure 2 ijerph-19-07292-f002:**
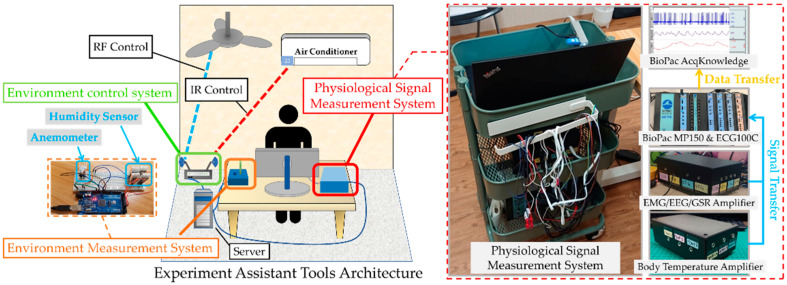
Experiment assistant tools.

**Figure 3 ijerph-19-07292-f003:**
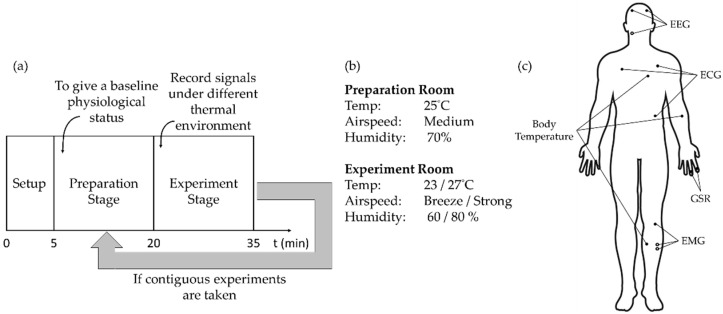
Experiment setups: (**a**) experiment procedure, (**b**) environment settings, and (**c**) measured signals and its body parts.

**Figure 4 ijerph-19-07292-f004:**
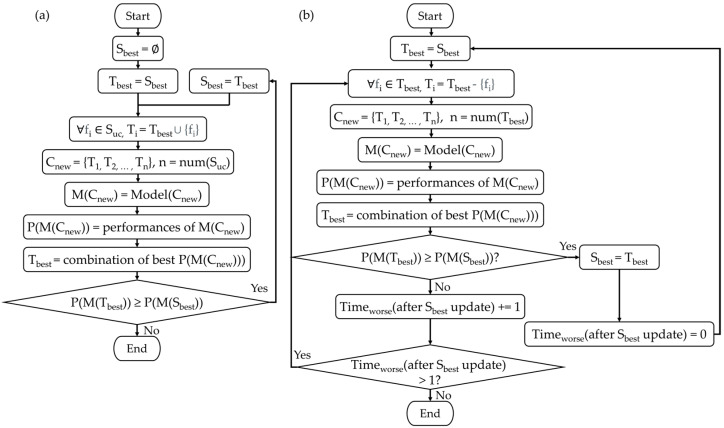
The modeling algorithm workflow. (**a**) Stage 1: feature increment. (**b**) Stage 2: redundant feature removal. The meaning of the symbols are as follows: S_best_: the feature combination of the best performed model. T_best_: the feature combination of the stepwise best performed model. S_uc_: the set of all unselected features. f_i_: symbol of a feature. C_new_: a set of all feature combinations in each step. M (combinations): model of the combinations. P (model): performance of the model. Time_worse_ (after S_best_ update): the time that new models perform worse than the current best model on record.

**Table 1 ijerph-19-07292-t001:** Extracted features and explanations.

Type	Source Signal	Feature	Explanation
Physiological	ECG	ECG_HR	Heart rate
		ECG_SDNN	The deviation of heart beat RR interval
		ECG_TP	ECG total power
		ECG_LF	ECG low-frequency band (0.04~0.15 Hz) relative power
		ECG_HF	ECG high-frequency band (0.15~0.4 Hz) relative power
		ECG_LF/HF	Ratio of ECG_LF and ECG_HF
	EMG	EMG_IEMG	Integration of EMG signal
		EMG_MAV	Mean absolute value of EMG
		EMG_RMS	Root mean square of EMG signal
		EMG_SSI	Simple square integration of EMG signal
	EEG	EEG_alpha	Alpha band average of EEG
		EEG_beta	Beta band average of EEG
		EEG_AVG	Average of EEG signal
		EEG_alpha_power	Relative power of EEG alpha band
		EEG_beta_power	Relative power of EEG beta band
	GSR	GSR_avg5 hz	Average of noise removed GSR signal (<5 hz)
	Body Temp.	T1 (Chest)	Average body temperature from chest
		T2 (Forearm)	Average body temperature from forearm
		T3 (Calf)	Average body temperature from calf
Environmental	Air Temp.	EnvTemp	Average air temperature
	Airspeed	EnvWind	Average air velocity
	Humidity	EnvRH	Average relative humidity

ECG: electrocardiography, EMG: electromyography, EEG: electroencephalography, GSR: galvanic skin response, Temp: temperature.

**Table 2 ijerph-19-07292-t002:** Correlation coefficient between thermal sensation and each of the features.

Type	Feature	Correlation Coefficient (r)	Significance (*p*-Value)
Physiological	ECG_HR	0.148	0.057
	ECG_SDNN *	−0.168	0.03
	ECG_TP	0.031	0.688
	ECG_LF	−0.006	0.941
	ECG_HF	0.029	0.704
	ECG_LF/HF	−0.049	0.53
	EMG_IEMG **	−0.215	0.005
	EMG_MAV **	−0.214	0.006
	EMG_RMS **	−0.216	0.005
	EMG_SSI **	−0.217	0.005
	EEG_alpha	0.055	0.475
	EEG_beta **	0.411	0
	EEG_AVG **	0.393	0
	EEG_alpha_power	−0.112	0.147
	EEG_beta_power	−0.112	0.147
	GSR_avg5 hz **	−0.285	0
	T1 (Chest) **	0.55	0
	T2 (Forearm) **	0.388	0
	T3 (Calf) **	0.558	0
Environmental	EnvTemp **	0.496	0
	EnvWind *	0.187	0.016
	EnvRH	−0.149	0.55

*. Correlation is significant at the 0.05 level (2-tailed); **. correlation is significant at the 0.01 level (2-tailed).

**Table 3 ijerph-19-07292-t003:** The best performed physiological feature combination.

Physiological Signal	Feature	Model RMSE	Model R^2^
EMG	EMG_MAV, EMG_IEMG, EMG_RMS	0.807	0.75
ECG	ECG_LF/HF, ECG_SDNN		
EEG	EEG_beta_power, EEG_beta		
Body Temp.	T3 (Calf), T2 (Chest)		

ECG: electrocardiography, EMG: electromyography, EEG: electroencephalography, Temp: temperature, RMSE: root-mean-square error, R^2^: R-squared.

**Table 4 ijerph-19-07292-t004:** Comparisons between the proposed model and PMV model.

	Proposed Model Performance		PMV Model Performance	
Trial Number	RMSE	R^2^	RMSE	R^2^
1	0.82	0.77	1.26	0.1
2	0.65	0.81	0.9	0.68
3	0.78	0.72	1.05	0.52
Average	0.75	0.77	1.07	0.43

RMSE: root-mean-square error, lower is better; R^2^: R-squared, higher is better.

**Table 5 ijerph-19-07292-t005:** Result of independent sample *t*-test.

	Mean (SD)	t	*p*-Value (2-Tailed)
Male	−0.21 (1.13)	−1.19	0.251
Female	0.40 (1.02)		

SD: standard deviation.

**Table 6 ijerph-19-07292-t006:** Result of Mann–Whitney U test.

	Median (Q1–Q3)	Z	*p*-Value (2-Tailed)
Male	−0.35 (−0.97–0.71)	−1.07	0.286
Female	0.31 (−0.52–1.33)		

Q1: first quartile, Q3: third quartile.

**Table 7 ijerph-19-07292-t007:** Sequence of ranking physiological signals.

Number of Physiological Signal	List of Signal	RMSE	R^2^
0	Environment Signal	1.03	0.58
1	Environment Signal, Body Temperature	0.97	0.63
1	Environment Signal, EMG	0.91	0.68
2	Environment Signal, EMG, Body Temperature	0.87	0.7
3	Environment Signal, EMG, Body Temperature, ECG	0.84	0.73
4	Environment Signal, EMG, Body Temperature, ECG, EEG	0.807	0.75

ECG: electrocardiography, EMG: electromyography, EEG: electroencephalography; RMSE: root-mean-square error, lower is better, R^2^: R-squared, higher is better.

## Data Availability

The data presented in this study are available on request from the corresponding author.

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
