# Peer review of "A Physiological-Signal-Based Thermal Sensation Model for Indoor Environment Thermal Comfort Evaluation"

_ijerph, 2022, doi:10.3390/ijerph19127292_

Round 1
Reviewer 1 Report
Table 5 needs a strong correct.
In my opinion, the authors did not consider several basic factors. The physiology of women and men is different, so the analysis done together may be misleading.
In addition, there is no information whether the people ate a meal before the test, what exactly the scope of duties they performed, which may affect the course of changes in body temperature, EEG, ECG, etc. It is important in which room the examined persons stayed - the area, arrangement of desks in relation to air conditioning, size of window openings. If a subject is struck by solar radiation, their perception may change regardless of the operating cooling system.
Is the 15-minute experiment sufficient to eliminate the subjects' emotions related to the study and give a reliable result? Is this time too short?
Reviewer 2 Report
Abstract: The key objective and problem statement with concluding remarks should be written.
Introduction: Need to add some recent references.
Methodology: Appropriate
Discussion: Add optimistic comparison with different related studies.
Conclusion: Should be add some future directions related to industrial implications.
In your discussion section, please link your empirical results with a broader and deeper literature review.
Please make sure your conclusions' section underscores the scientific value-added of your paper, and/or the applicability of your findings/results.
Highlight the novelty of your study. In addition to summarising the actions taken and results, please strengthen the explanation of their significance. It is recommended to use quantitative reasoning comparing with appropriate benchmarks, especially those stemming from previous work.
Please consult the journal's reference style for the exact appearance of these elements, and use of punctuation and capitalisation. Bibliography style is not always consistent, please check the reference section carefully and correct the inconsistency.
Please eliminate those multiple references. After that please check the manuscript thoroughly and eliminate ALL the lumps in the manuscript. This should be done by characterising each reference individually. This can be done by mentioning 1 or 2 phrases per reference to show how it is different from the others and why it deserves mentioning. Please eliminate the use of redundant words. Eg. In this way, Recently, Respectively, therefore, currently, thus, hence, finally, to do this, first, in order, however, moreover, nowadays, today, consequently, in addition, additionally, on the other hand, furthermore. –
Please revise all similar cases, as removing these term(s) would not significantly affect the meaning of the sentence. This will keep the manuscript as CONCISE as possible. Please check ALL.
Avoid beginning or end a sentence with one or a few words, they are usually redundant. E.g. Today,.Avoid beginning a sentence with a conjunction term, e.g. And, Which, Where, Because.
